# Quantitative Monitoring of Mycelial Growth of *Aspergillus fumigatus* in Liquid Culture by Optical Density

Ken Miyazawa,[a] Takashi Umeyama,[a] Yasutaka Hoshino,[a] Keietsu Abe,[b] Yoshitsugu Miyazaki[a]

aDepartment of Fungal Infection, National Institute of Infectious Diseases, Tokyo, Japan
bLaboratory of Applied Microbiology, Department of Microbial Biotechnology, Graduate School of Agricultural Science, Tohoku University, Sendai, Japan

**ABSTRACT** Filamentous fungi form multicellular hyphae, which generally form pellets in liquid shake cultures, during the vegetative growth stage. Because of these characteristics, growth-monitoring methods commonly used in bacteria and yeast have not been applied to filamentous fungi. We have recently revealed that the cell wall polysaccharide $\alpha$-1,3-glucan and extracellular polysaccharide galactosaminogalactan (GAG) contribute to hyphal aggregation in *Aspergillus oryzae*. Here, we tested whether *Aspergillus fumigatus* shows dispersed growth in liquid media that can be quantitatively monitored, similar to that of yeasts. We constructed a double disruptant mutant of both the primary $\alpha$-1,3-glucan synthase gene *ags1* and the putative GAG synthase gene *gtb3* in *A. fumigatus* AfS35 and found that the hyphae of this mutant were fully dispersed. Although the mutant lost $\alpha$-1,3-glucan and GAG, its growth and susceptibility to antifungal agents were not different from those of the parental strain. Mycelial weight of the mutant in shake-flask cultures was proportional to optical density for at least 18 h. We were also able to quantify the dose response of hyphal growth to antifungal agents by measuring optical density. Overall, we established a convenient strategy to monitor *A. fumigatus* hyphal growth. Our method can be directly used for screening for novel antifungals against *Aspergillus* species.

**IMPORTANCE** Filamentous fungi generally form hyphal pellets in liquid culture. This property prevents filamentous fungi so that we may apply the methods used for unicellular organisms such as yeast and bacteria. In the present study, by using the fungal pathogen *Aspergillus fumigatus* strain with modified hyphal surface polysaccharides, we succeeded in monitoring the hyphal growth quantitatively by optical density. The principle of this easy measurement by optical density could lead to a novel standard of hyphal quantification such as those that have been used for yeasts and bacteria. Dose response of hyphal growth by antifungal agents could also be monitored. This method could be useful for screening for novel antifungal reagents against *Aspergillus* species.

**KEYWORDS** filamentous fungi, hyphal aggregation, optical density, $\alpha$-1,3-glucan, galactosaminogalactan, *Aspergillus fumigatus*

Growth of bacteria or yeast can be easily quantified, and their cultures can be used for susceptibility testing. However, because filamentous fungi form mycelia, limited methods have been developed to monitor their growth. Conventionally, growth of filamentous fungi has been assessed by measuring dry or wet biomass (1). The former approach is very accurate but time-consuming because it requires tens of milligrams of dried fungi (1). The latter approach is fast, but results in considerable variation for samples of the same dry weight (1). To precisely quantify fungal mass, several direct or indirect measurement methods have been developed. The mass of fungi grown in wood is quantified by chitin content (2). The amount of ergosterol, a unique component of fungal cells, is also useful (3, 4). Quantitative PCR can be used to quantify fungal cells in soil and infected hosts. Banerjee et al. have measured the turbidity

Address correspondence to Yoshitsugu Miyazaki, ym46@niid.go.jp.

The authors declare no conflict of interest.

of ground hyphal cells (5). A method for measurement of fluorescence of formazan produced by living cells from 2,3-bis-(2-methoxy-4-nitro-5-sulfophenyl)-tetrazolium-5-carboxanilide (XTT) has been developed (6). Recently, several methods based on image analysis have been reported (7–10).

In filamentous fungi, the surface structure differs between conidia and hyphae. The outer layer of conidia, termed the rodlet layer, is composed of polymerized hydrophobin (11, 12) and is underlaid by melanin (13). Below the melanin is the cell wall composed mainly of polysaccharides (i.e., glucan, chitin, and mannan) (13, 14). When hyphae extend from conidia, the polysaccharide layer is exposed to the surface, and the structure of the polysaccharide network is continuously modified (15). In *Aspergillus* species, hyphae have $\alpha$-glucan in the outermost layer, which covers the $\beta$-1,3-glucan and chitin layers (16). In the hyphal growth stage, filamentous fungi secrete extracellular matrix (ECM) composed mainly of polysaccharides, proteins, and lipids (13, 17). In *Aspergillus* species, the main ECM polysaccharides are galactosaminogalactan (GAG), $\alpha$-glucan, and galactomannan (18).

In shake-flask cultures, extended hyphae of filamentous fungi sometimes form pellets, which compromise the accuracy of growth monitoring. During hyphal growth, specific interactions of conidial wall components have been thought to be the primary cause of aggregation (19, 20), but the specific component has been revealed only recently. Fontaine et al. have reported that $\alpha$-1,3-glucan directly contributes to aggregation of germinating conidia in *Aspergillus fumigatus* (21). We have reported that $\alpha$-1,3-glucan is an adhesive factor for hyphae in the model fungus *Aspergillus nidulans* and the industrial fungus *Aspergillus oryzae* (22–24), and He et al. have reported $\alpha$-1,3-glucan as a hyphal adhesive factor in *A. nidulans* (25). We have revealed that GAG contributes to hyphal aggregation in *A. oryzae* and that the hyphae of a strain deficient in both $\alpha$-1,3-glucan and GAG are dispersed in shake-flask cultures (26).

In the present study, we studied aggregation of *A. fumigatus*, a conditional pathogen of medical importance. We constructed a double disruptant of *ags1* and *gtb3*, which encode putative primary $\alpha$-1,3-glucan and GAG synthases, respectively, and observed that the hyphae of this *A. fumigatus* mutant were dispersed. We monitored the growth of the mutant by measuring optical density at 600 nm ($OD_{600}$) in three different media. We also assessed whether the value of $OD_{600}$ is directly proportional to the mycelial weight. We compared minimal inhibitory concentrations (MICs) determined by $OD_{600}$ measurements and by the standard Clinical and Laboratory Standards Institute (CLSI) antifungal susceptibility testing.

## RESULTS

**The double disruptant of ags1 and gtb3 of *A. fumigatus* showed dispersed hyphae in shake-flask culture.** *A. fumigatus* possesses three $\alpha$-1,3-glucan synthase genes, *ags1*, *ags2*, and *ags3*, and *ags1* has a primary role in $\alpha$-1,3-glucan biosynthesis (27–29). Here, we constructed an *ags1* disruption strain of *A. fumigatus* and examined its macromorphology in shake-flask culture. Hyphal pellets were smaller in the Δ*ags1* strain than in the AfS35 strain (Fig. 1A), suggesting that $\alpha$-1,3-glucan contributes to hyphal aggregation in *A. fumigatus*. We also constructed a strain with disruption of the *gtb3* gene, which encodes putative GAG synthase (30). Although the pellet morphology of the Δ*gtb3* strain was similar to that of AfS35 (Fig. 1A), it did not form a biofilm (Fig. S4A). To generate a double mutant, the *ags1* and *gtb3* genes were sequentially disrupted (Δ*ags1*Δ*gtb3*); the hyphae of the Δ*ags1*Δ*gtb3* strain were completely dispersed in shake-flask culture (Fig. 1A), consistent with the *A. oryzae* mutant deficient in $\alpha$-1,3-glucan and GAG (26). All the mutants constructed here showed similar radial growth and conidiation on agar plates (Fig. S4B). The antifungal susceptibilities of Δ*ags1*Δ*gtb3* and AfS35 were similar (Table 1).

To characterize the disruption strains of *ags1* and/or *gtb3*, we evaluated colony growth under different temperature, osmotic, pH, and cell wall stresses. At each of the temperatures tested (30, 37, 42, and 45°C), different strains showed almost the same radial growth (Fig. S5A). The colonies of the Δ*ags1*Δ*gtb3* strain were light brown, whereas those of the other strains were green at 42 and 45°C (Fig. S5A). Under osmotic

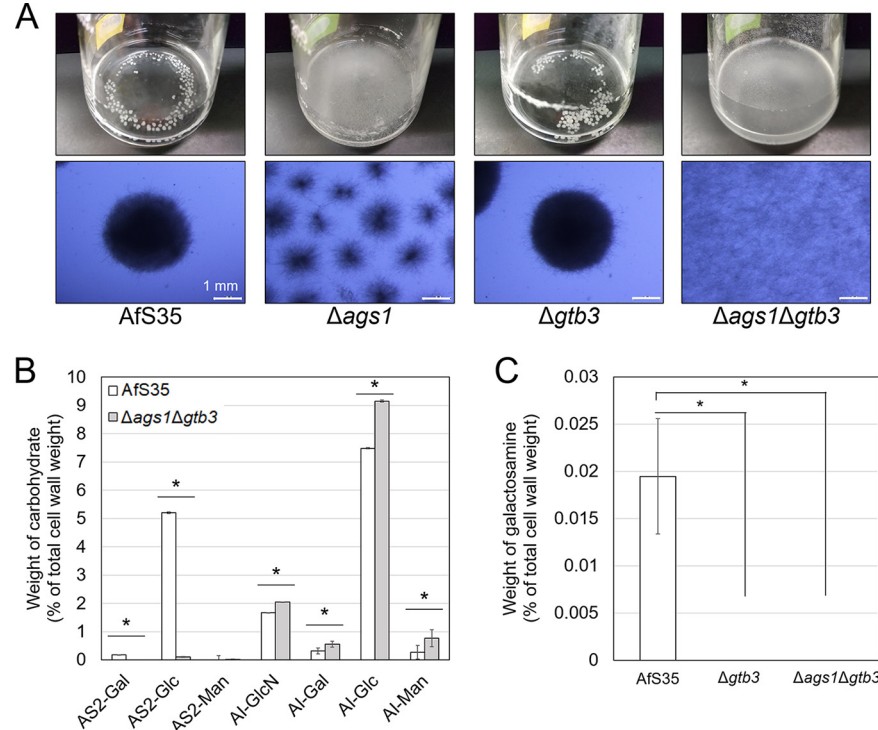

**FIG 1** Characterization of the Δ*ags1*, Δ*gtb3*, and double-disruptant strains of *A. fumigatus*. (A) Growth characteristics of each strain. Conidia (5.0 × 10⁵/mL) of AfS35 (control) or mutant strains were inoculated into liquid *Aspergillus* minimal medium (AMM) and rotated at 160 rpm at 37°C for 24 h. (B) Monosaccharide composition of cell wall AS2 and AI fractions from the AfS35 and Δ*ags1*Δ*gtb3* strains. Error bars show the standard deviation from three replicates. *, $P < 0.05$ (Student's *t* test). (C) Galactosamine content in the hot-water-soluble fraction from the AfS35, Δ*gtb3*, and Δ*ags1*Δ*gtb3* strains. *, $P < 0.05$ (Tukey's test). AI, alkali-insoluble fraction; AS2, alkali-soluble fraction; Glc, glucose; GlcN, glucosamine; Gal, galactose; Man, mannose.

stress, the growth of all the strains was slower on media containing 1 M NaCl or 1 M KCl than on the other media but did not differ considerably among the four strains (Fig. S5B). All the strains showed similar growth at pH 4, 7, and 8 (Fig. S5C). On *Aspergillus* minimal medium containing 0.1% yeast extract (YAMM) containing Congo red, the Δ*ags1* and Δ*ags1*Δ*gtb3* strains did not grow at 10 μg/mL, and only AfS35 grew at 20 μg/mL (Fig. S5D). On YAMM containing 20 μg/mL calcofluor white, AfS35 grew weakly, and the other strains did not grow (Fig. S5D). Overall, growth was scarcely different among the strains except in the presence of cell wall stress-inducing reagents.

To validate that the hyphal dispersion of Δ*ags1*Δ*gtb3* was caused by a lack of α-1,3-glucan and GAG, we analyzed the cell wall components. The alkali-soluble (AS2) fraction mainly contains α-1,3-glucan with a small amount of galactomannan, and the alkali-insoluble (AI) fraction contains β-1,3-glucan, chitin, and galactomannan (29, 31).

**TABLE 1** Antifungal susceptibility determined using the Clinical and Laboratory Standards Institute M38-A2 method[a]

| Strain | MIC[b] or MEC[c] value (μg/mL) | | | | | | | |
|---|---|---|---|---|---|---|---|---|
| | MCFG | CPFG | AMB | 5FC | FLC | ITC | VRC | MCZ |
| AfS35 | ≤0.015 | 0.25 | 0.5 | >64 | >64 | 0.5 | 2 | 8 |
| Δ*ags1*Δ*gtb3* | ≤0.015 | 0.5 | 0.5 | >64 | >64 | 0.5 | 1 | 8 |

[a]MCFG, micafungin; CPFG, caspofungin; AMB, amphotericin B; 5FC, flucytosine; FLC, fluconazole; ITC, itraconazole; VRC, voriconazole; MCZ, miconazole.
[b]Lowest concentration of the drug (other than MCFG and CPFG) that prevents any discernible growth (100% inhibition).
[c]Lowest concentrations of MCFG and CPFG that lead to growth of small, rounded, compact hyphal forms.

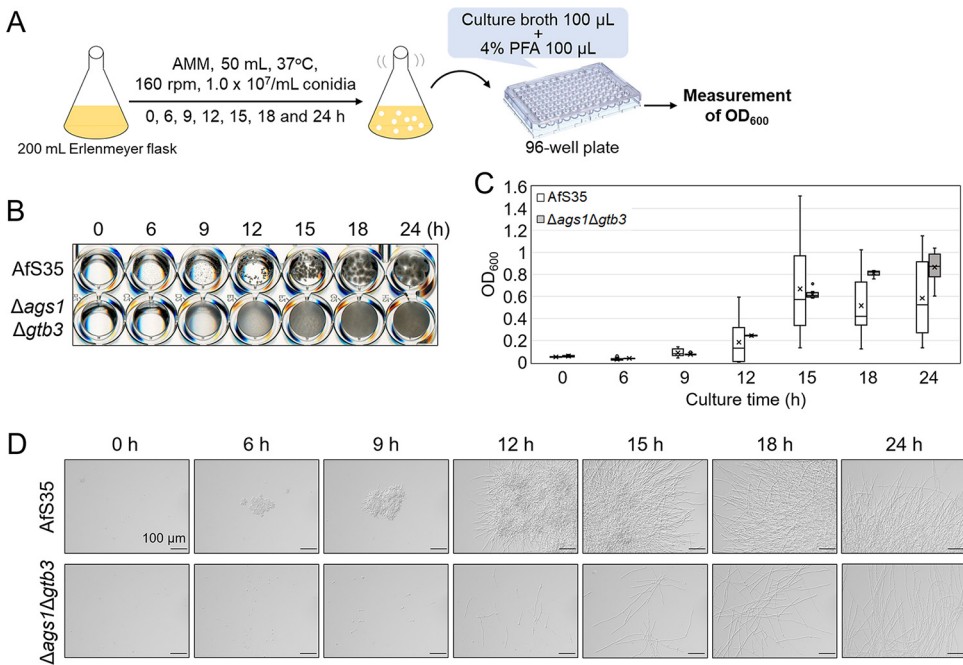

**FIG 2** Evaluation of growth of the AfS35 and Δ*ags1*Δ*gtb3* strains by optical density. (A) Scheme of the experiment. Conidia were inoculated into AMM liquid medium, and aliquots were withdrawn at the indicated time points. The culture broth was mixed with 4% paraformaldehyde (PFA) in a 96-well plate, and the optical density at 600 nm (OD$_{600}$) values were measured with a microplate reader. (B) Growth of the AfS35 and Δ*ags1*Δ*gtb3* strains. (C) Time course of OD$_{600}$. The OD$_{600}$ values were calculated from 12 measurements per time point and are shown as box plots. The lines in the boxes indicate medians, and the crosses indicate averages. The circles indicate outliers. (D) Time course of hyphal morphology of the AfS35 and Δ*ags1*Δ*gtb3* strains.

The AS2 fraction from the AfS35 strain contained 5.2% of glucose (AS2-Glc), and the AI fraction contained 1.7% of glucosamine (AI-GlcN) and 7.5% of glucose (AI-Glc) (Fig. 1B). The minor components (<1% each) were galactose (AS2-Gal and AI-Gal) and mannose (AI-Man) (Fig. 1B). Galactosamine was scarcely detected in the AS2 and AI fractions from the AfS35 strain. The Δ*ags1*Δ*gtb3* strain contained only 0.1% of AS2-Glc, which was significantly less than in AfS35 ($P < 0.05$; Fig. 1B), suggesting that α-1,3-glucan was almost abolished by disruption of the *ags1* gene. The hot-water-soluble (HW) fraction from AfS35 contained 0.019% of galactosamine, whereas galactosamine was scarcely detected in the HW fraction from the Δ*ags1*Δ*gtb3* strain (Fig. 1C). Galactosamine was almost abolished in the HW fraction from the Δ*gtb3* strain (Fig. 1C). These results suggest that disruption of the *gtb3* gene resulted in a complete loss of GAG. The contents of AI-GlcN, AI-Gal, AI-Glc, and AI-Man were slightly higher in the Δ*ags1*Δ*gtb3* strain than in AfS35 (Fig. 1B), which might be attributable to compensation of the defect in α-1,3-glucan and/or GAG in the Δ*ags1*Δ*gtb3* strain.

**Growth of the Δ*ags1*Δ*gtb3* strain can be monitored by optical density at 600 nm.** Dispersed hyphal morphology of the Δ*ags1*Δ*gtb3* strain prompted us to try to monitor its growth by optical density. Conidia of the AfS35 and Δ*ags1*Δ*gtb3* strains were inoculated into 50 mL of liquid *Aspergillus* minimal medium (AMM) and cultured with shaking at 37°C for up to 24 h. At the indicated time points, the culture was dispensed into a 96-well plate and fixed, and OD$_{600}$ was measured (Fig. 2A). Pellets became visible in AfS35 from 9 h, and their size increased with time, whereas the Δ*ags1*Δ*gtb3* hyphae were continuously dispersed (Fig. 2B). The OD$_{600}$ measurements of AfS35 showed that, for example, the first and third quartiles from 12 measurements were 0.894 and 0.337, respectively, at 15 h (Fig. 2C), suggesting no correlation between apparent AfS35 growth and OD$_{600}$. In the Δ*ags1*Δ*gtb3* strain, the first and third quartiles were 0.633 and 0.595 at 15 h (Fig. 2C), suggesting that measurement of OD$_{600}$ is suitable for monitoring the growth of the mutant. The OD$_{600}$ measurements seemed to

be reproducible at least until 18 h in the Δ*ags1*Δ*gtb3* strain (Fig. 2C). Microscopic observations showed that AfS35 formed aggregates as early as during conidial swelling at 6 h, whereas pellet formation was scarce in the Δ*ags1*Δ*gtb3* strain up to 24 h (Fig. 2D). Although the growth rate was different in RPMI and yeast extract-glucose (YG) media, the growth of the Δ*ags1*Δ*gtb3* strain could also be monitored by $OD_{600}$ values (Fig. S6A and B). At 18 h of culture, the Δ*ags1*Δ*gtb3* strain formed visible pellets in YG medium (Fig. S6A).

To evaluate the linearity of $OD_{600}$ values versus mycelial weight, a dilution series of Δ*ags1*Δ*gtb3* culture was prepared, and $OD_{600}$ was measured. A plot of mycelial weight versus the reciprocal of dilution rate showed that the dilution was performed accurately (Fig. S7A). The relationship between $OD_{600}$ values and the reciprocal of dilution rate was linear below $OD_{600} = 0.75$ (Fig. S7B). Using a standard curve, mycelial weight could be determined from $OD_{600}$ values (Fig. S7C). Overall, growth of the Δ*ags1*Δ*gtb3* strain in shake-flask culture could be easily and precisely monitored by measuring culture optical density.

We further investigated whether growth monitoring by optical density would be applicable to the *A. oryzae* strain deficient in both α-1,3-glucan and GAG (AGΔ-GAGΔ) (Fig. S8A). The hyphae of AGΔ-GAGΔ were dispersed in AMM, RPMI, and YG media, whereas the wild-type strain formed pellets (Fig. S8B to D). The growth of AGΔ-GAGΔ was monitored by $OD_{600}$ (Fig. S8B to D). These results suggest that monitoring of the hyphal growth by optical density is also applicable to *A. oryzae*.

To reveal why Δ*ags1*Δ*gtb3* formed hyphal pellets in YG medium, we labeled α-1,3-glucan with α-1,3-glucan-binding domain fused to green fluorescent protein (AGBD-GFP). In the AfS35 strain cultured in YG medium, α-1,3-glucan was clearly labeled along the outline of the cells (Fig. S9). In Δ*ags1*Δ*gtb3* cultured in YG medium, AGBD-GFP fluorescence was observed in the septa and the outline of the cells, especially in the apical region (Fig. S9). AMM-cultured Δ*ags1*Δ*gtb3* was scarcely labeled with AGBD-GFP (Fig. S9). The hyphae of *A. oryzae* WT cultured in YG were clearly labeled, whereas those of *A. oryzae* AGΔ-GAGΔ cultured in either AMM or YG were scarcely labeled (Fig. S9).

**Antifungal susceptibility of the Δ*ags1*Δ*gtb3* strain evaluated from optical density measurements.** As an application of the monitoring method, we evaluated the dose response of growth to the presence of antifungal agents: amphotericin B (AMB), flucytosine (5FC), itraconazole (ITC), voriconazole (VRC), and micafungin (MCFG). The Δ*ags1*Δ*gtb3* strain was grown in RPMI medium containing an antifungal agent for 15 h in a 48-well plate, and then $OD_{600}$ was measured. Growth was repressed in the presence of ITC and VRC in a dose-dependent manner (Fig. 3). Relative growth rate was around 10% in spite of high drug concentrations because of the presence of conidia in each well. Conidia were swollen but did not germinate in the presence of high concentrations of the drugs (data not shown). Drug concentrations that caused 50% inhibition of growth were 0.125 μg/mL for VRC and 0.0313 μg/mL for ITC (Fig. 3). Growth was completely inhibited at 2 μg/mL of VRC and 1 μg/mL of ITC (Fig. 3), which was in agreement with MICs determined by CLSI M38-A2 (Table 1). The concentrations that caused 50% growth inhibition were 0.0625 μg/mL for AMB and 4 μg/mL for 5FC (Fig. 3). Growth was completely inhibited at 0.5 μg/mL of AMB and 256 μg/mL of 5FC (Fig. 3). Growth inhibition by MCFG was hard to evaluate (Fig. 3) because this drug had a limited effect in the early stage of germination. Overall, the growth of the Δ*ags1*Δ*gtb3* strains could be quantified by optical density in a 48-well plate in the presence of the antifungal agents tested except for MCFG.

## DISCUSSION

In the present study, we constructed a double disruptant of the *ags1* and *gtb3* genes, which have roles in α-1,3-glucan and GAG biosynthesis, respectively, and observed that the hyphae of the mutant were fully dispersed in shake-flask culture. Using this mutant, we accurately monitored hyphal growth by measuring optical density for the first time in filamentous fungi and evaluated dose-dependent growth in the presence of antifungal agents.

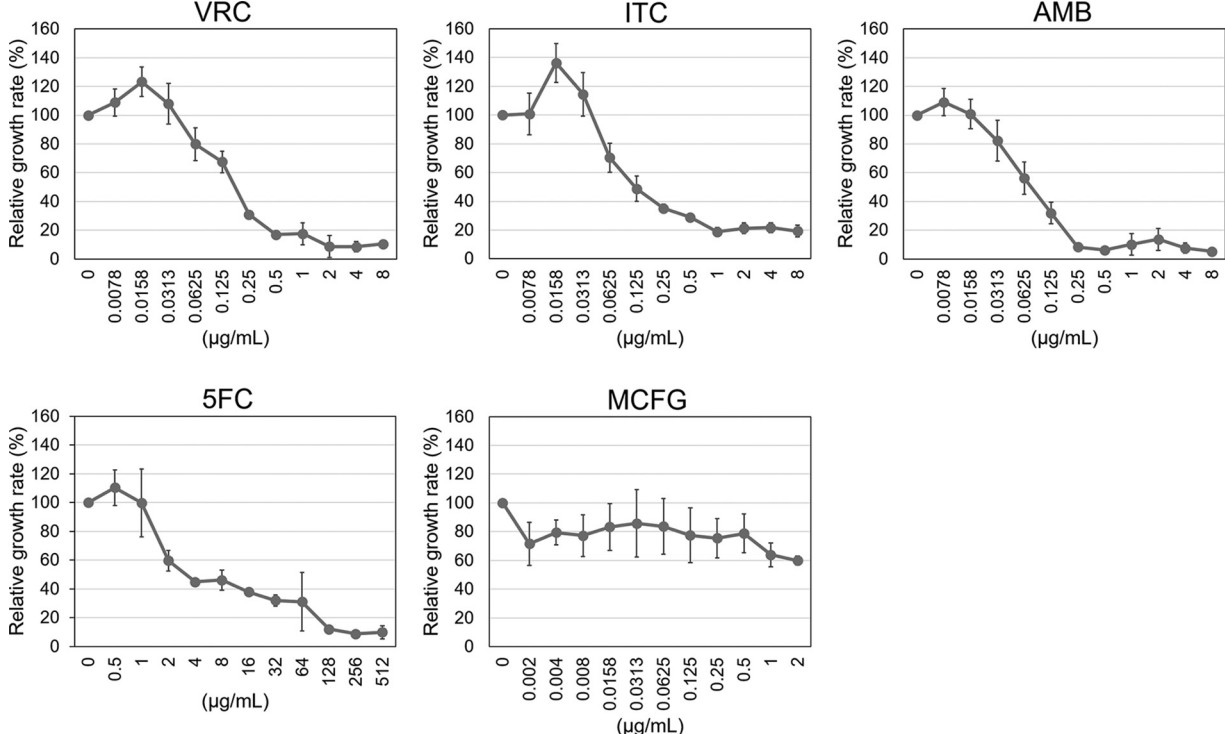

**FIG 3** Growth of the Δ*ags1*Δ*gtb3* strain in the presence of antifungal agents monitored by $OD_{600}$. Conidia ($5.0 \times 10^6$/mL) were inoculated into 500 $\mu$l of RPMI liquid medium containing the indicated antifungal agent in a 48-well plate and rotated at 300 rpm at 35°C for 15 h. The $OD_{600}$ values were measured with a microplate reader. Error bars show the standard deviations from three biological replicates. AMB, amphotericin B; 5FC, flucytosine; ITC, itraconazole; VRC, voriconazole; MCFG, micafungin.

The growth of filamentous fungi is conventionally monitored by measuring dry mass; it is easy and precise but requires tens of milligrams of material, so precise quantification of cells during the early stage of germination is difficult. The use of optical density has allowed us to quantify limited cell numbers of a filamentous fungus.

Biosynthesis of GAG is thought to be regulated by a cluster of five genes: *uge3*, *sph3*, *ega3*, *agd3*, and *gtb3* in *A. fumigatus* (32). Disruption of the *uge3* and *sph3* genes results in the absence of GAG in *A. fumigatus* (33, 34). The *gtb3* disruptant constructed here showed no biofilm formation (Fig. S4A). GalN was hardly detected in the *gtb3* disruptant (Fig. 1C), suggesting that Gtb3 is essential for GAG biosynthesis. Although Gtb3 seems to be involved in synthesizing polymers of galactose and *N*-acetylgalactosamine from UDP-galactose and UDP-*N*-acetylgalactosamine (30, 35), direct biochemical evidence has not been reported. To unveil the mechanism of GAG biosynthesis, further biochemical analyses are needed.

Hyphal pellets in Δ*ags1* that are smaller than those in the AfS35 strain of *A. fumigatus* (Fig. 1A) support the role of $\alpha$-1,3-glucan in hyphal aggregation and are consistent with the phenotypes of $\alpha$-1,3-glucan-deficient strains of *A. oryzae*, *A. nidulans*, and *Aspergillus niger* (22, 23, 25, 36, 37). In the Δ*ags1*Δ*gtb3* strain, AS2-Glc was scarcely detected (Fig. 1B). Beauvais et al. reported that an *A. fumigatus* mutant with the disrupted *ags1* gene lost half of the amount of cell wall $\alpha$-1,3-glucan of the parental strain (27); this difference might be attributable to the difference between the parental strains.

In the Δ*ags1*Δ*gtb3* strain, the contents of AI-GlcN, AI-Gal, AI-Glc, and AI-Man were increased (Fig. 1B). In *A. fumigatus*, a defect in $\alpha$-1,3-glucan or $\beta$-1,3-glucan is compensated by other cell wall polysaccharides (27, 29, 38); some glucan and chitin synthases of fungi use UDP-hexose and UDP-N-acetylhexosamine as the substrates (39, 40). In *A. nidulans* and *A. oryzae*, deficiency in cell wall $\alpha$-1,3-glucan scarcely affects other cell wall polysaccharides (22–25, 36), suggesting that the compensation machinery is specific to *A. fumigatus* among *Aspergillus* species.

We monitored the growth of the Δ*ags1*Δ*gtb3* strain in AMM, RPMI, or YG medium (Fig. 2; Fig. S6). In AMM, mycelial weight started increasing at 9 to 12 h; this increase flattened out at 18 to 24 h; and the increase per unit time (i.e., slope of the line connecting data points) was greatest at 12 to 15 h (Fig. 2C). The growth was reproducible in AMM for at least 18 h (Fig. 2C) and in RPMI for 24 h, when the culture reached stationary phase (Fig. S6). In YG medium, the growth was reproducible for up to 12 h, but the variability increased by 15 h (Fig. S6B), and hyphal pellets were formed at 18 h (Fig. S6A). When AGBD-GFP was used to label $\alpha$-1,3-glucan in the pellets of Δ*ags1*Δ*gtb3* cultured for 24 h in YG medium, GFP fluorescence was detected in the septa and along the outline of the cells (Fig. S9). The hyphae of Δ*ags1*Δ*gtb3* cultured for 24 h in AMM medium were almost fully dispersed and were scarcely labeled with GFP (Fig. S9). Because *A. fumigatus* has three $\alpha$-1,3-glucan synthase genes (*ags1*, *ags2*, and *ags3*), *ags2* and/or *ags3* might be specifically expressed in YG medium and contribute to pellet formation. Although the diameter of *A. oryzae* conidia is larger than that of *A. fumigatus* conidia, we were able to quantitatively monitor the growth of *A. oryzae* AGΔ-GAGΔ in AMM, RPMI, and YG media (Fig. S8). Overall, AMM and RPMI appear to be the best media for quantitative evaluation of the growth of *A. fumigatus* Δ*ags1*Δ*gtb3*.

As an application of our monitoring method for antifungal susceptibility testing, we cultured Δ*ags1*Δ*gtb3* in RPMI medium containing an antifungal agent for 15 h in a 48-well plate and were able to quantify dose-dependent growth (Fig. 3). The concentration that completely inhibited growth was in agreement with MICs determined by CLSI M38-A2 (Fig. 3). We propose that conditions for liquid culture according to CLSI M38-A2 are appropriate for this purpose.

Monitoring growth by optical density is superior to that by conventional methods for several reasons: (i) growth monitoring is quantitative and continuous. During drug testing based on CLSI M38-A2, growth has to be observed visually. The mutant with dispersed hyphae would allow establishment of automated drug screening for *Aspergillus*. (ii) Fungicidal and fungistatic drugs could be selected using our strategy. We propose to screen anti-*Aspergillus* drugs from a drug library, although drugs that do not inhibit germination but disorder hyphal extension, such as echinocandin, might be hard to select using our method. Recently, Beattie and Krysan (41) reported that the release of intracellular adenylate kinase from hyphal cells is a sensitive readout to detect cell lysis and is useful for screening antifungal reagents against *A. fumigatus*. In combination with the adenylate kinase method, our strategy may allow selection of anti-*Aspergillus* drugs with various spectra by monitoring optical density of fungal culture.

Fungi, especially filamentous fungi, are phenotypically heterogeneous in their growth (42). When filamentous fungi form pellets in liquid culture, oxygen reaches only 200 $\mu$m from the pellet surface (43). Therefore, cellular conditions seem to differ considerably between the surface and the interior of the pellet. The hyphae of Δ*ags1*Δ*gtb3* are easily dispersed in liquid culture and thus seem to have relatively constant cellular metabolism, although metabolic differences between apical and subapical cells of hyphae are hardly avoidable. Dispersed hyphae of Δ*ags1*Δ*gtb3* in culture could be useful to analyze cellular responses such as autophagy, metabolic changes in the presence of antifungal agents, and responses to alteration of growth conditions.

We quantified the growth of *A. fumigatus* in the presence of antifungal agents by $OD_{600}$ using the strain with dispersed hyphae. This strain could be used as a model for the high-throughput screening of antifungal compounds. However, a limitation of our study is that a gene-deletion strain has to be used. The decrease in growth in the presence of an antifungal agent depended on its type, but the mutant hyphae remained dispersed. Thus, changes in gene expression induced by antifungal agents should be more clearly observed in the mutant than in the parental strain, which forms hyphal pellets. Specific inhibitors of $\alpha$-1,3-glucan and GAG synthesis have not been identified. Screening for such inhibitors could contribute to establishing the cultivation of dispersed hyphae to perform experiments similar to those reported here in clinical isolates without gene disruption. As another approach, genome editing could accelerate

**TABLE 2** Strains used in this study

| Species | Strain | Genotype | Reference |
|---|---|---|---|
| A. fumigatus | AfS35 | akuA::loxP | (44) |
| | Δags1 | akuA::loxP, ags1::hph | This study |
| | Δgtb3 | akuA::loxP, gtb3::hph | This study |
| | Δags1 (loxP) | akuA::loxP, ags1::lox72 | This study |
| | Δags1Δgtb3 | akuA::loxP, ags1::lox72, gtb3::hph | This study |
| A. oryzae | Wild-type | ΔligD::sC, ΔadeA::ptrA, niaD⁻, adeA⁺ | (52) |
| | AGΔ-GAGΔ | ΔligD::sC, ΔadeA::ptrA, niaD⁻, adeA⁺, agsA::loxP, agsB::loxP, agsC::loxP, sphZugeZ::adeA | (26) |

generation of strains deficient in $\alpha$-1,3-glucan and GAG from clinically isolated strains. Cell sorting might be used to isolate single germinating conidia of *A. fumigatus* that are resistant to some antifungal agents. We believe that dispersion of hyphae could dramatically extend the applicability of analytical methods for filamentous fungi.

## MATERIALS AND METHODS

**Strains and growth media.** Strains used in this study are listed in Table 2. The nonhomologous end-joining-deficient strain of *A. fumigatus*, AfS35, was used for all genetic manipulations (44). *Aspergillus* minimal medium (AMM) (45) was used for harvesting conidia and liquid cultivation of *A. fumigatus* strains. Czapek-Dox (CD) medium (Becton, Dickinson and Company, Sparks, NV) was used for transformation of *A. fumigatus*. For cultivation of *A. oryzae* wild-type and AGΔ-GAGΔ strains in AMM, 70 mM sodium nitrite was used as a nitrogen source instead of sodium nitrate. YG medium (46) was used for liquid cultivation in some experiments. RPMI 1640 medium buffered with morpholinepropanesulfonic acid (MOPS; 0.165 M; pH 7.0) was used for shake-flask and plate cultures to evaluate susceptibility to antifungal agents.

**Evaluation of growth under stress conditions.** For evaluation of growth on agar plates under stress conditions, YAMM was used (47). For osmotic stress, 1 M NaCl, 1 M KCl, or 1.2 M sorbitol was added. For cell wall stress, calcofluor white (fluorescent brightener 28) (MP Biomedicals, Irvine, CA) or Congo red (Fujifilm Wako Pure Chemical Corp., Osaka, Japan) were added from 100-fold concentrated stock solutions. The response to temperature was evaluated at 30, 37, 42, and 45°C. Acid or alkaline stress was evaluated at pH 4 or 8. The medium pH was adjusted with HCl or NaOH.

**Construction of plasmids and strains.** The sequences of all primers are listed in Table S1. Transformation of *A. fumigatus* was performed as described previously (48) with some modifications. Briefly, conidia were inoculated into YG medium (10 mL) and incubated at 37°C with shaking at 150 rpm. Conidial suspension was mixed with 10 mL of KCl-citric acid solution (1.1 M KCl, 100 mM potassium citrate buffer, pH 5.8) containing 0.4 g of VinoTaste Pro (Novozymes, Bagsværd, Denmark) and incubated for 1 h at 30°C with gentle shaking. Generated protoplasts were transformed with 2 to 5 $\mu$g of DNA cassettes, plated onto CD medium supplemented with 1 M sucrose, and incubated for 15 h at 37°C. Then CD medium containing 0.35% Bacto Agar and 400 $\mu$g/mL hygromycin was overlaid onto the CD plate.

**Construction of the Δags1 strain.** The first round of PCR-amplified gene fragments contained the 5′ noncoding region (amplicon 1), the coding region (amplicon 2) of *ags1* from AfS35 genomic DNA, and the *hph* gene (amplicon 3) from pSK397 (44) (Fig. S1A). Amplicon 1 was amplified with primers Afags1-LU and Afags1-LL, amplicon 2 was amplified with primers Afags1-RU and Afags1-RL, and amplicon 3 was amplified with primers 397-5 and 397-3. The three resulting PCR products were gel-purified and fused into a disruption cassette in the second round of PCR. The resulting PCR product was gel-purified and transformed into the AfS35 strain (Fig. S1B). Replacement of the *ags1* gene was confirmed by PCR (Fig. S1C).

**Construction of the Δags1 (loxP) strain.** The *ags1* gene was disrupted by using the Cre/loxP marker recycling system (49). The plasmid pAH-Cre was first constructed (Fig. S2A) as follows. The *hph* marker (amplicon 1) was amplified from pSK397. A fragment containing the *lox71*, *xynG2* promoter (P*xynG2*), and *Cre* (amplicon 2) was amplified from the plasmid pAAG-Cre (49). A fragment containing *ampR* and *ori* (amplicon 3) was amplified from pUC19. Amplicon 1 was amplified with primers IF1-lox66-hph-Fw and 397-5, amplicon 2 was amplified with primers IF2-PxynG2-Fw and IF2-lox71-TagdA-Rv, and amplicon 3 was amplified with primers IF3-lox71-pUC19 and IF3-lox66-pUC19. The three amplicons were fused by using a NEBuilder HiFi DNA assembly kit (New England Biolabs, Ipswich, MA). The pAH-Cre plasmid was used as a template for PCR with primers M13-47 and RV-M, resulting in amplicon C. PCR was performed to amplify gene fragments containing the 5′ noncoding region (amplicon L) and the coding region (amplicon R) of *ags1* from AfS35 genomic DNA. Amplicon L was amplified with primers Afags1-LU and Afags1-LL-M13, and amplicon R with primers Afags1-RU-M13 and Afags1-RL. Amplicons C, L, and R were gel-purified and fused into a disruption cassette by PCR (Fig. S2B). The resulting PCR product was gel-purified and transformed into the AfS35 strain (Fig. S2C). Candidate strains were selected on AMM medium containing 400 $\mu$g/mL hygromycin and then cultured on AMM medium without hygromycin containing 1% xylose to induce *Cre* expression (Fig. S2C). Strains sensitive to hygromycin were isolated by culture on AMM plates with or without hygromycin. Replacement of the *ags1* gene was confirmed by PCR (Fig. S2D).

**Disruption of the *gtb3* gene to generate single (Δ*gtb3*) and double (Δ*ags1*Δ*gtb3*) mutants.** The first round of PCR-amplified gene fragments contained the 5′ noncoding region (amplicon 1), the coding region (amplicon 2) of *gtb3* from AfS35 genomic DNA, and the *hph* gene (amplicon 3) from pSK397 (Fig. S3A). Amplicon 1 was amplified with primers Afgtb3-LU and Afgtb3-LL, amplicon 2 was amplified with primers Afgtb3-RU and Afgtb3-RL, and amplicon 3 was amplified with primers 397-5 and 397-3. The three resulting PCR products were gel-purified and fused into a disruption cassette in the second round of PCR. The resulting PCR product was gel-purified and transformed into the AfS35 strain to generate Δ*gtb3* or into the Δ*ags1* (*loxP*) strain to generate Δ*ags1*Δ*gtb3* (Fig. S3B). Replacement of the *gtb3* gene was confirmed by PCR (Fig. S3C).

**Biofilm visualization.** Conidia (final concentration, $1 \times 10^6$/mL) of the AfS35, Δ*ags1*, Δ*gtb3*, and Δ*ags1*Δ*gtb3* strains were inoculated into 1 mL of AMM medium in a polystyrene 24-well plate and incubated for 24 h at 37°C. The culture medium was removed, the plate was washed twice with phosphate-buffered saline (PBS), and 0.5 mL of 0.5% (wt/vol) crystal violet solution was added. The plate was incubated at room temperature for 5 min. Excess stain was removed, and the plate was washed twice with water. The biofilm was imaged with a flatbed scanner (GT-X970; Seiko Epson Corp., Nagano, Japan).

**Determination of cell wall components.** Mycelia were cultured for 24 h in AMM medium, collected by filtering through Miracloth (Merck Millipore, Darmstadt, Germany), washed twice with water, and freeze-dried. Cell wall components were fractionated as described previously with some modifications (22). The mycelia were ground with a mortar and pestle, and the powder (1 g) was suspended in 40 mL of 0.1 M sodium phosphate buffer (pH 7.0). The mycelial suspension was autoclaved at 121°C for 60 min and centrifuged at $10,000 \times g$ for 10 min. The supernatant was retained, and the pellet was resuspended in 40 mL of 0.1 M phosphate buffer (pH 7.0), autoclaved, and centrifuged again at $10,000 \times g$ for 10 min. The supernatants from the first two centrifugations were combined, dialyzed against water, and lyophilized, resulting in the HW fraction. The pellet was suspended in 50 mL of 0.1 M NaOH for 6 h at 4°C. The suspension was centrifuged at $10,000 \times g$ for 10 min, and the pellet was suspended in 50 mL of 0.1 M NaOH and centrifuged again at $10,000 \times g$ for 10 min. The supernatant was designated the diluted alkali-soluble fraction. The pellet was suspended in 50 mL of 2 M NaOH for 24 h at 4°C and centrifuged as above. The supernatant and the pellet were designated the alkali-soluble (AS) and alkali-insoluble (AI) fractions, respectively. These fractions were neutralized with acetic acid and dialyzed against water, and the AS fraction was centrifuged. The supernatant was designated AS1, and the precipitate was designated AS2. All the fractions were freeze-dried and weighed. Monosaccharide composition of the HW, AS2, and AI fractions was quantified according to Yoshimi et al. (22).

***In vitro* antifungal susceptibility testing.** MICs and minimal effective concentrations (MECs) of AfS35 and Δ*ags1*Δ*gtb3* strains against micafungin (MCFG), caspofungin (CPFG), amphotericin B (AMB), flucytosine (5FC), fluconazole (FLC), itraconazole (ITC), voriconazole (VRC), and miconazole (MCZ) were evaluated using a frozen plate for antifungal susceptibility testing of yeasts (Eiken Chemicals, Tokyo, Japan) according to the method for CLSI M38-A2 (50).

**Measurement of optical density.** To measure the optical density of conidia and mycelia in shake-flask culture, conidia (final concentration, $1.0 \times 10^7$/mL) of AfS35 or Δ*ags1*Δ*gtb3* strain were inoculated into 50 mL of AMM, YG, or RPMI medium in a 200-mL Erlenmeyer flask and rotated at 160 rpm at 37°C. At each sampling point, the culture (2 mL) was withdrawn with wide-bore tips, and 100 $\mu$l of the culture was mixed by pipetting with wide-bore tips with 100 $\mu$l of 100 mM sodium phosphate buffer (pH 7.0) containing 4% paraformaldehyde in a 96-well plate. $OD_{600}$ was measured in a microplate reader (Synergy LX, BioTek, Winooski, VT). The morphology of conidia and mycelia was observed under an IX81 inverted fluorescence microscope (Olympus, Tokyo, Japan). For evaluation of *A. oryzae* growth, conidia (final concentration, $1.0 \times 10^7$/mL) of the wild-type or AGΔ-GAGΔ strain were inoculated into 50 mL of AMM, YG, or RPMI medium in a 200-mL Erlenmeyer flask and rotated at 160 rpm at 35°C, and $OD_{600}$ was measured as for *A. fumigatus*.

To measure the optical density of the dilution series of mycelial suspensions, conidia (final concentration, $1.0 \times 10^7$/mL) of the Δ*ags1*Δ*gtb3* strain were inoculated into 50 mL of AMM medium in a 200-mL Erlenmeyer flask and rotated at 160 rpm at 37°C for 18 h. The culture was filtered through Miracloth, and the mycelia were resuspended in 10 mL of water; 5 mL of the suspension was mixed with 5 mL of water (2-fold dilution). By repeating this procedure, the dilution series were prepared; 200 $\mu$l of each dilution was dispensed into 5 wells of a 96-well plate, and $OD_{600}$ was measured. The remaining mycelial suspension (4 mL) was freeze-dried and weighed.

**Labeling of $\alpha$-1,3-glucan with AGBD-GFP.** Mycelial cells fixed with paraformaldehyde were placed on a glass slide and washed twice with PBS. A drop of PBS containing 100 $\mu$g/mL AGBD-GFP (51) was added, the sample was incubated at room temperature for 30 min, washed as above, and imaged on the IX81.

**Evaluation of antifungal susceptibility of the Δ*ags1*Δ*gtb3* strain by optical density.** To evaluate the susceptibility of the Δ*ags1*Δ*gtb3* strain to MCFG, AMB, 5FC, ITC, and VRC, conidia (final concentration, $5.0 \times 10^6$/mL) were inoculated into 500 $\mu$l of RPMI medium containing an antifungal agent in a 48-well plate and rotated at 300 rpm using MicroMixer E-36 (Taitec, Koshigaya, Japan) at 35°C for 15 h, and $OD_{600}$ was measured in triplicate. Each test was performed in triplicate, and standard deviations were determined. The relative growth rate was calculated as the percentage of $OD_{600}$ at each drug concentration relative to the mean $OD_{600}$ in the absence of the drug.

MCFG, AMB, 5FC, ITC, and VRC were prepared according to CLSI M38-A2 (50). Briefly, water-insoluble AMB, VRC, and ITC were dissolved in DMSO, and then the dilution series were prepared by mixing with DMSO. Water-soluble MCFG and 5FC were dissolved in RPMI medium, and the dilution series were prepared by mixing with RPMI.

## SUPPLEMENTAL MATERIAL

Supplemental material is available online only.

**SUPPLEMENTAL FILE 1**, PDF file, 1.2 MB.

## ACKNOWLEDGMENTS

We thank Shigekazu Yano (Yamagata University) for kindly providing AGBD-GFP.

This work was supported by Japan Society for the Promotion of Science (JSPS) KAKENHI grants JP20K22773, JP20H02895, and JP20K08834. This study was partly supported by the Joint Usage/Research Program of Medical Mycology Research Center, Chiba University (20-5), and by Japan Agency for Medical Research and Development (AMED) under grants JP21fk0108135 and JP21fk0108094.

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
