## [Reviewer comments · Microbiology Spectrum]

Microbiology Spectrum

Quantitative monitoring of mycelial growth of *Aspergillus fumigatus* in liquid culture by optical density

Ken Miyazawa, Takashi Umeyama, Yasutaka Hoshino, Keietsu Abe, and Yoshitsugu Miyazaki

Corresponding Author(s): Yoshitsugu Miyazaki, National Institute of Infectious Diseases

Review Timeline:

Submission Date:	April 29, 2021
Editorial Decision:	June 10, 2021
Revision Received:	September 21, 2021
Editorial Decision:	October 4, 2021
Revision Received:	November 15, 2021
Accepted:	November 17, 2021

Editor: Slavena Vylkova

Reviewer(s): Disclosure of reviewer identity is with reference to reviewer comments included in decision letter(s). The following individuals involved in review of your submission have agreed to reveal their identity: Xiaoxiao He (Reviewer #1); Takuji Oka (Reviewer #3)

Transaction Report:

DOI: <https://doi.org/10.1128/Spectrum.00063-21>

June 10, 2021

Dr. Yoshitsugu Miyazaki
National Institute of Infectious Diseases
Department of Chemotherapy and Mycoses
Tokyo 162-8640
Japan

Re: Spectrum00063-21 (Quantitative monitoring of mycelial growth of *Aspergillus fumigatus* with modified surface polysaccharides in liquid culture)

Dear Dr. Yoshitsugu Miyazaki:

Thank you for submitting your manuscript to Microbiology Spectrum. As you will see the reviewers support publication of a revised paper. Please revise the paper along the lines suggested by the reviewers. Given that this manuscript was initially submitted to another ASM journal with a different focus, I encourage you to specifically address the outlined points to reflect the scope of Microbiology.

1. The authors provide a different strategy for rapid screening of various compounds that can alter growth or viability of *A. fumigatus*. The method relies on use of mutant strain with severely altered cell wall composition. Thus, it becomes unclear how well use of this strain will reflect the wild type phenotypes, especially given the heterogeneity within the species. Reviewer #2 specifically noted that the discussion is rather short and it is advisable to add such explanation.

2. I especially agree with Reviewer #1 comments 2 and 5. As this appears as a technique manuscript a clear and well detailed explanation of all experimental procedures, materials used, etc. is needed. This would ensure the reproducibility of the presented approach. Regarding comment 5 either a direct comparison between solid and liquid medium growth and justification for benefits (growth rate, ease of experimental setup, low costs...) have to be included. I strongly recommend such paired solid-liquid growth comparison between wt and mutant using a handful of conditions as the goal of the study is to demonstrate wide usability of the presented approach.

Of note, problems with the file containing comments made by reviewer 2 were noted. I have attached the correct file as an attachment.

When submitting the revised version of your paper, please provide (1) point-by-point responses to the issues raised by the reviewers as file type "Response to Reviewers," not in your cover letter, and (2) a PDF file that indicates the changes from the original submission (by highlighting or underlining the changes) as file type "Marked Up Manuscript - For Review Only". Please use this link to submit your revised manuscript - we strongly recommend that you submit your paper within the next 60 days or reach out to me. Detailed information on submitting your revised paper are below.

Link Not Available

Sincerely,

Slavena Vylkova

Journals Department
Reviewer comments:

Reviewer #1 (Comments for the Author):

In this manuscript, Ken and colleagues constructed a mutant strain of *A. fumigatus* that lacks the alpha-glucan and galactosaminogalactan in the cell wall. Based on this mutant strain, they developed a strategy to assess the cell growth by measuring optical density. As the authors proposed, this new strain may be applied for high throughput anti-fungal drug screening in shaken liquid growth condition. However, there are several concerns for the high throughput screening strategy. Substantially more experiment should be performed before it could be validated strategy

Major points:

1, Growth of unicellular organism and filamentous fungi (multicellular) are fundamentally different. Use of optical density for measuring the growth of unicellular organism is based on their uniform distribution. In contrast, filamentous fungi grow in a way to elongate the existing mycelia, as the authors show in Fig.2D. The conidia and mycelia of filamentous fungi tend to cluster together and form pellets in a shaken liquid culture. Even the Δ ags1 Δ gtb3 mutant strain, the mycelia were not totally separated from each other. They still form small but visible pellets in the shaken liquid medium, especially in the YG medium. If so, it is hard to agree with the concept that a simple measurement of optical density could faithfully represent the cell density.

2, Materials and Methods section. For the measurement of optical density, authors described that "100 μ L was mixed with 100 μ L of 100 mM sodium phosphate buffer (pH 7.0) containing 4% paraformaldehyde in a 96-well plate". However, this statement was not clear. Especially when the pellets of the colonies become larger in late time points, the transfer of such colony suspension cannot be performed by simple pipetting (not sure if it is a problem for Δ ags1 Δ gtb3 mutant strain, but it must be a problem for a wildtype strain). Authors should add more details for these steps. Otherwise the result could be quite inconsistent due to different operations.

3, Regarding to the data consistency, optical density measurements were performed in different medium cultures, such as AMM, YG and RPMI. The time course of OD600 results were shown in Fig2C and FigS5B. Comparing these data, the growth rate was different in each medium and data reproducibility was also quite different. Especially in YG medium the data reproducibility was the worst. As Fig. S5A showed, Δ ags1 Δ gtb3 mutant strain formed more visible pellets in YG medium than that in RPMI and AMM. Authors should compare such data and draw a conclusion for the best growth condition (medium selection and time for growth) of the proposed strategy. Otherwise, such variations will greatly limit the use of this strategy.

4, The strategy that described in this manuscript based on a Δ ags1 Δ gtb3 mutant strain. As the authors and many other publications noted, such mutant strain has very different cell wall architecture, which means it responds differently as a wildtype strain. In addition, mutant *A. fumigatus* strains that lack alpha-glucan and GAG are both less virulent. Therefore, it would be hard to tell if the outcomes from this strategy could also be useful for the clinical important strains, which should not be alpha-glucan and GAG defective strains.

5 Other strategies have also be used for testing drug sensitivity of filamentous fungi, such as testing colony growth on drug containing solid medium, or testing the pellets diameter in a shaken liquid culture. To validate the application of the proposed strategy in this manuscript, authors should compare these different methods and discuss how the proposed method may overwhelm other methods.

Minor point:

The title of the manuscript was not straightforward to represent the work in this study.

line 208, gene names are not italic

there is a typo in line 322, "he" should be "the"

Did author test the GAG content in gtb3 single deletion strain? It would be very helpful to show that gtb3 directly regulated GAG formation in *A. fumigatus*.

In their previous work, authors have generated other alpha-glucan and GAG defective *A. nidulans* and *A. oryzae* strains. Authors may consider to test their strategy with more strains to validate this idea.

Reviewer #2 (Comments for the Author):

incorporate all the comments indicated in the manuscript.

Staff Comments:

Preparing Revision Guidelines

For complete guidelines on revision requirements, please see the Instructions to Authors at [link to page]. **Submissions of a paper that does not conform to Microbiology Spectrum guidelines will delay acceptance of your manuscript.**

Please return the manuscript within 60 days; if you cannot complete the modification within this time period, please contact me. If you do not wish to modify the manuscript and prefer to submit it to another journal, please notify me of your decision immediately so that the manuscript may be formally withdrawn from consideration by Microbiology Spectrum.

If you would like to submit an image for consideration as the Featured Image for an issue, please contact Spectrum staff.

**Quantitative monitoring of mycelial growth of *Aspergillus fumigatus* with modified**
**surface polysaccharides in liquid culture**

Ken Miyazawa^a, Takashi Umeyama^a, Yasutaka Hoshino^a, Keietsu Abe^b, and Yoshitsugu
Miyazaki^a

7 ^aDepartment of Fungal Infection, National Institute of Infectious Diseases, Tokyo, Japan

8 ^bLaboratory of Applied Microbiology, Department of Microbial Biotechnology,
Graduate School of Agricultural Science, Tohoku University, Sendai, Japan

Keywords: filamentous fungi, hyphal aggregation, optical density, α -1,3-glucan,
galactosaminogalactan

Address correspondence to Y. Miyazaki, ym46@niid.go.jp

**Running title** (< 54 characters)

Quantification of *A. fumigatus* by optical density

[revised manuscript text omitted]

Unlike in unicellular microorganisms including yeasts, the colony forming unit
values in filamentous fungi cannot be determined from turbidity. Growth of filamentous
fungi is conventionally monitored using methods that have limitations such as
measurement of dry mass (1). The dry mass method is easy and precise, but it requires
tens of milligrams of material and is time consuming (1). Because of the biomass
requirement, precise quantification of cells during the early stage of germination is
particularly difficult. Here, we used turbidity to quantify a limited amount of cells of a
filamentous fungus. In AMM, biomass started increasing at 9–12 h and this increase
flattened out at 18–24 h, and the increase per unit time (i.e., slope of the line connecting
data points) was greatest at 12–15 h (Fig. 2C).

Biosynthesis of GAG is thought to be regulated by a cluster of five genes (*uge3*,

*sph3*, *ega3*, *agd3*, and *gtb3* in *A. fumigatus*) (40). Disruption of the *uge3* and *sph3* genes
results in the absence of GAG in *A. fumigatus* (41, 42). The *gtb3* disruptant constructed
here showed no biofilm formation (Fig. S4A). GalN was hardly detected in the *gtb3*
disruptant (Fig. 1C), suggesting that Gtb3 is essential for GAG biosynthesis. Although
Gtb3 seems to be involved in synthesizing polymers of galactose and
*N*-acetylgalactosamine from UDP-galactose and UDP-*N*-acetylgalactosamine (38, 43),
direct biochemical evidence has not been reported. To unveil the mechanism of GAG
biosynthesis, further biochemical analyses are needed.

Here, we quantified the growth of *A. fumigatus* in the presence of antifungal
agents by OD₆₀₀ using the strain with dispersed hyphae. This strain could be used as a
model for the high-throughput screening of antifungal compounds. However, our study
has a limitation that has to use of a gene-deletion strain. The decrease in growth in the
presence of an antifungal agent depended on its type, but the mutant hyphae remained
dispersed in spite of the agent. These results suggest that the gene expression alteration
induced by antifungal agents should be more clearly observed in the mutant than in the
parental strain, which forms hyphal pellets. Genome-editing can accelerate generation
of a strain that deficient in α -1,3-glucan and GAG from clinically isolated strain. Cell
sorting might be used to isolate single germinating conidia of *A. fumigatus* that are

resistant to some antifungal agents. We believe that dispersion of cells could
dramatically extend the applicability of the analytical methods for filamentous fungi.

**Acknowledgements**

[revised manuscript text omitted]

fraction from the AfS35 and $\Delta ags1\Delta gtb3$ strains.

**Figure 2. Evaluation of growth of the AfS35 and $\Delta ags1\Delta gtb3$ strains by turbidity.**
(A) Scheme of the experiment. Conidia were inoculated into AMM liquid medium and
aliquots were withdrawn at the indicated time points. The culture broth was mixed with
4% paraformaldehyde (PFA) in a 96-well plate, and the OD₆₀₀ values were measured
with a microplate reader. (B) Growth of the AfS35 and $\Delta ags1\Delta gtb3$ strains. (C) Time
course of OD₆₀₀. The OD₆₀₀ values were calculated from 12 measurements per time
point and are shown as box plots. Lines in boxes indicate medians, and crosses indicate
averages. Circles indicate outliers. (D) Time course of hyphal morphology of the AfS35
and $\Delta ags1\Delta gtb3$ strains.

**Figure 3. Growth of the $\Delta ags1\Delta gtb3$ strain in the presence of antifungal agents**

**monitored by OD₆₀₀.** Conidia (5.0×10^6 /mL) were inoculated into 500 μ L of RPMI
liquid medium containing the indicated antifungal agent in a 48-well plate and rotated at
300 rpm at 35°C for 15 h. The OD₆₀₀ values were measured with a microplate reader.
Error bars are standard deviations from three biological replicates.

Responses to Reviewers

Reviewer #1

In this manuscript, Ken and colleagues constructed a mutant strain of *A. fumigatus* that lacks the alpha-glucan and galactosaminogalactan in the cell wall. Based on this mutant strain, they developed a strategy to assess the cell growth by measuring optical density. As the authors proposed, this new strain may be applied for high throughput anti-fungal drug screening in shaken liquid growth condition. However, there are several concerns for the high throughput screening strategy. Substantially more experiment should be performed before it could be validated strategy.

Major points:

1, Growth of unicellular organism and filamentous fungi (multicellular) are fundamentally different. Use of optical density for measuring the growth of unicellular organism is based on their uniform distribution. In contrast, filamentous fungi grow in a way to elongate the existing mycelia, as the authors show in Fig. 2D. The conidia and mycelia of filamentous fungi tend to cluster together and form pellets in a shaken liquid culture. Even the $\Delta ags1\Delta gtb3$ mutant strain, the mycelia were not totally separated from each other. They still form small but visible pellets in the shaken liquid medium, especially in the YG medium. If so, it is hard to agree with the concept that a simple measurement of optical density could faithfully represent the cell density.

Our experiments suggest that AMM and RPMI medium are suitable for the evaluation of the growth of the *A. fumigatus* $\Delta ags1\Delta gtb3$ strain by optical density. To gain insight into why small but visible pellets were formed in YG medium, we labelled the $\Delta ags1\Delta gtb3$ mutant grown in YG medium for 24 h with alpha-1,3-glucanase

alpha-1,3-glucan-binding domain (AGBD) fused with GFP (AGBD-GFP), and observed clear labelling of the septa and the outline of the cell. As *A. fumigatus* has three alpha-1,3-glucan synthase genes (*ags1—3*), alpha-1,3-glucan in labeled hyphae of the $\Delta ags1\Delta gtb3$ mutant was likely synthesized by Ags2 and/or Ags3. In *A. oryzae* $\Delta agsA\Delta agsB\Delta agsC\Delta sphZ\Delta ugeZ$ (AG Δ -GAG Δ) all the three genes encoding alpha-1,3-glucan synthases were disrupted, and this mutant had fully dispersed hyphae in all the media tested including YG medium. Although the data on *A. oryzae* AG Δ -GAG Δ are consistent with the above data on *A. fumigatus* $\Delta ags1\Delta gtb3$, further experiments are needed to prove the effect of alpha-1,3-glucan on pellet formation in the $\Delta ags1\Delta gtb3$ mutant. We have revised the manuscript in lines 354—370.

2, Materials and Methods section. For the measurement of optical density, authors described that "100 μ L was mixed with 100 μ L of 100 mM sodium phosphate buffer (pH 7.0) containing 4% paraformaldehyde in a 96-well plate". However, this statement was not clear. Especially when the pellets of the colonies become larger in late time points, the transfer of such colony suspension cannot be performed by simple pipetting (not sure if it is a problem for $\Delta ags1\Delta gtb3$ mutant strain, but it must be a problem for a wildtype strain). Authors should add more details for these steps. Otherwise the result could be quite inconsistent due to different operations.

We have added more details as follows.

Line 196—199: The culture (2 mL) was withdrawn with wide-bore tips at each sampling point, and 100 μ L of the culture was mixed by pipetting with wide-bore tips with 100 μ L of 100 mM sodium phosphate buffer (pH 7.0) containing 4% paraformaldehyde in a 96-well plate.

3, Regarding to the data consistency, optical density measurements were performed in

different medium cultures, such as AMM, YG and RPMI. The time course of OD600 results were shown in Fig2C and FigS5B. Comparing these data, the growth rate was different in each medium and data reproducibility was also quite different. Especially in YG medium the data reproducibility was the worst. As Fig. S5A showed, $\Delta ags1\Delta gtb3$ mutant strain formed more visible pellets in YG medium than that in RPMI and AMM. Authors should compare such data and draw a conclusion for the best growth condition (medium selection and time for growth) of the proposed strategy. Otherwise, such variations will greatly limit the use of this strategy.

We have revised the text to describe the best conditions for the proposed strategy in lines 354—376.

4, The strategy that described in this manuscript based on a $\Delta ags1\Delta gtb3$ mutant strain. As the authors and many other publications noted, such mutant strain has very different cell wall architecture, which means it responds differently as a wildtype strain. In addition, mutant *A. fumigatus* strains that lack alpha-glucan and GAG are both less virulent. Therefore, it would be hard to tell if the outcomes from this strategy could also be useful for the clinical important strains, which should not be alpha-glucan and GAG defective strains.

We have explained the outcomes of this strategy as follows.

Line 390—399: Fungi, especially filamentous fungi, are phenotypically heterogeneous in their growth. When filamentous fungi form pellets in liquid culture, oxygen reaches only 200 μm from the pellet surface. Therefore, cellular conditions seem to differ considerably between the surface and the interior of the pellet. The hyphae of $\Delta ags1\Delta gtb3$ are easily dispersed in liquid culture and thus seem to have relatively

constant cellular metabolism, although metabolic differences between apical and subapical cells of hyphae are hardly avoidable. Dispersed hyphae of $\Delta ags1\Delta gtb3$ in culture could be useful to analyze cellular responses such as autophagy, metabolic changes in the presence of antifungal agents, and responses to alteration of growth conditions.

At present, we are investigating culture conditions that prevent pellet formation in the parental strain, which would be useful for testing clinical isolates. The perspectives are described in the manuscript as follows.

Line 407—410: Establishing culture conditions that prevent pellet formation of a strain with an intact cell wall structure could expand the application of growth monitoring by optical density. Understanding the biochemical and physicochemical properties of α -1,3-glucan and GAG will contribute to finding suitable culture conditions.

5 Other strategies have also be used for testing drug sensitivity of filamentous fungi, such as testing colony growth on drug containing solid medium, or testing the pellets diameter in a shaken liquid culture. To validate the application of the proposed strategy in this manuscript, authors should compare these different methods and discuss how the proposed method may overwhelm other methods.

We have revised the manuscript to compare our method with conventional methods as follows.

Line 377—389: Monitoring growth by optical density is superior to that by conventional methods for several reasons: 1) growth monitoring is quantitative and continuous. During drug testing based on CLSI M38-A2, growth has to be observed visually. The mutant with dispersed hyphae would allow establishment of automated drug screening for *Aspergillus*. 2) Fungicidal and fungistatic drugs could be selected

using our strategy. We propose to screen anti-*Aspergillus* drugs from a drug library, although drugs that do not inhibit germination but disorder hyphal extension, such as echinocandin, might be hard to select using our method. Recently, Beattie and Krysan reported that the release of intracellular adenylate kinase from hyphal cells is a sensitive readout to detect cell lysis and is useful for screening antifungal reagents against *A. fumigatus*. In combination with the adenylate kinase method, our strategy may allow selection of anti-*Aspergillus* drugs with various spectra by monitoring optical density of fungal culture.

Minor point:

1 The title of the manuscript was not straightforward to represent the work in this study.

We have changed the title as follows:

Quantitative monitoring of mycelial growth of *Aspergillus fumigatus* in liquid culture by optical density

2 line 208, gene names are not italic

Plain text is correct in this case because the whole heading is in italics.

3 there is a typo in line 322, "he" should be "the"

Corrected.

4 Did author test the GAG content in *gtb3* single deletion strain? It would be very helpful to show that *gtb3* directly regulated GAG formation in *A. fumigatus*.

We have added the data on GalN content of Δ *gtb3* to Fig. 1C.

5 In their previous work, authors have generated other alpha-glucan and GAG defective *A. nidulans* and *A. oryzae* strains. Authors may consider to test their strategy with more strains to validate this idea.

We have monitored the growth of the *Aspergillus oryzae* mutant lacking both alpha-1,3-glucan and GAG, and added Figure S7. As expected, our strategy was applicable to this *A. oryzae* mutant. We have revised the text (lines 202–205 in the Materials and Methods, lines 292–297 in the Results, and Table 1). Unfortunately, the *Aspergillus nidulans* mutant is not available to me at my current institute.

Reviewer #2

Incorporate all the comments indicated in the manuscript.

Line 92: The abbreviation should be indicated in bracket as it is for the 1st time.

We have added the names of YG and RPMI media in full on lines 92–94.

Line 100: Why 37⁰?

To simply quantify growth, we cultured the fungi at 37°C. To evaluate drug sensitivity, we cultured them at 35°C according to the protocols of CLSI M38-A2.

Line 233: Why here? This should be in the discussion part unless otherwise the Results and Discussion parts merged together.

We have moved the sentence to the Discussion section (lines 340–343).

Line 256—259: More evidences from previous researchers that support the current finding should be required in the discussion part.

We have explained the supposed mechanism of the increase in AI-Glc and AI-GlcN in *Δags1Δgtb3* and cited appropriate references (lines 347–353).

Line 314: The discussion part is very short and not well expressed. In paragraph 1, 2 and 4 you only present the finding and not well interpreted and do not compared with previous findings. Generally the discussion part require major revision

We have substantially revised the Discussion.

October 4, 2021

Dr. Yoshitsugu Miyazaki
National Institute of Infectious Diseases
Department of Chemotherapy and Mycoses
Tokyo 162-8640
Japan

Re: Spectrum00063-21R1 (Quantitative monitoring of mycelial growth of *Aspergillus fumigatus* in liquid culture by optical density)

Dear Dr. Yoshitsugu Miyazaki:

Cell wall rearrangements were shown to lead to differences in antifungal susceptibility. Both reviewers suggested to perform additional tests using common cell wall stress agents and conditions (T, pH, etc.) or different medium. Furthermore, the α -1,3 beta glucan labeling with GFP should be added to the manuscript, as suggested by reviewer 1. Visualization of the labeling under different conditions (different medium or cell wall stress) would further strengthen the manuscript findings. Please include the requested modifications to the revised version.

Thank you for submitting your manuscript to Microbiology Spectrum. When submitting the revised version of your paper, please provide (1) point-by-point responses to the issues raised by the reviewers as file type "Response to Reviewers," not in your cover letter, and (2) a PDF file that indicates the changes from the original submission (by highlighting or underlining the changes) as file type "Marked Up Manuscript - For Review Only". Please use this link to submit your revised manuscript - we strongly recommend that you submit your paper within the next 60 days or reach out to me. Detailed information on submitting your revised paper are below.

Link Not Available

Sincerely,

Slavena Vylkova

Journals Department
Reviewer comments:

Reviewer #1 (Comments for the Author):

The authors have addressed most of my concerns during the revision. Especially the elaborated Discussion pointed out the potential advantages and drawbacks of the proposed strategy. I have only one concern left regarding the α -1,3-glucan compensation in YG medium. It is unclear why the result of α -1,3-glucan labeling with GFP was not shown in the manuscript. Author should add such data in the revised manuscript.

Moreover, if re-formation of α -1,3-glucan was the reason for the pellet in YG medium. This result suggested that other factors, especially the medium composition, may challenge the proposed strategy. Beauvais has previously generated a triple A. *fumigatus* deletion stain that had no α -1,3-glucan at all. And there should be no compensation of α -1,3-glucan in this strain. I would suggest the authors to further construct a new mutant stain based on this triple deletion strain. This would further warranty the compatibility of the proposed strategy.

Reviewer #3 (Comments for the Author):

First, I am an additional reviewer after the first review. The manuscript entitled "Quantitative monitoring of the mycelial growth of *Aspergillus fumigatus* in liquid culture by optical density" written by Miyazawa K et al. describes a method to monitor hyphal growth of pathogenic filamentous fungus, *A. fumigatus*, showing age1 and gtb3 double mutant using absorbance OD600 as an indicator. This is also the first study to demonstrate that Gtb3 is involved in biosynthesizing glycosaminoglycans (GAGs). The method was used to assess the effects of antifungal drugs used in clinical treatment, and the results were consistent with those based on the method described in CLSI M38-A2. Monitoring growth by absorbance allows for rapid screening of antifungal drugs. Additionally, this study was robustly conducted and had no technical problems. However, I think this study needs to be revised on several points before publication.

1. I feel uncomfortable with the choice of the word "biomass." I think the term is inappropriate to describe the tiny weight of fungus.
2. The cell wall of age1 and gtb3 double mutant may be thinner than the parental strain, making it easier for antifungal agents to penetrate it. Considering such influence, I think the number of antifungal agents' examples in the experiment is too small. Therefore, please present additional data that have been widely verified, such as the effects of chemical compounds (except the antifungal agents shown here) and the effects of some stress conditions (temperature, osmotic pressure and pH etc).
3. In the discussion section (line 407), you have said that "establishing culture conditions that prevent pellet formation helps screen antifungal drugs using clinical isolate strain," but this has nothing to do with your findings. Rather, if inhibitors of alpha-glucan and GAG biosynthesis are discovered, you could perform the similar experiment with the inhibitor on a clinical strain. Therefore, I think you should discuss this point.

Staff Comments:

Preparing Revision Guidelines

Please return the manuscript within 60 days; if you cannot complete the modification within this time period, please contact me. If you do not wish to modify the manuscript and prefer to submit it to another journal, please notify me of your decision immediately so that the manuscript may be formally withdrawn from consideration by Microbiology Spectrum.

The authors have addressed most of my concerns during the revision. Especially the elaborated Discussion pointed out the potential advantages and drawbacks of the proposed strategy. I have only one concern left regarding the α -1,3-glucan compensation in YG medium. It is unclear why the result of α -1,3-glucan labeling with GFP was not shown in the manuscript. Author should add such data in the revised manuscript.

Moreover, if re-formation of α -1,3-glucan was the reason for the pellet in YG medium. This result suggested that other factors, especially the medium composition, may challenge the proposed strategy. Beauvais has previously generated a triple *A. fumigatus* deletion strain that had no α -1,3-glucan at all. And there should be no compensation of α -1,3-glucan in this strain. I would suggest the authors to further construct a new mutant strain based on this triple deletion strain. This would further warranty the compatibility of the proposed strategy.

Response to reviewers

Reviewer #1 (Comments for the Author):

The authors have addressed most of my concerns during the revision. Especially the elaborated Discussion pointed out the potential advantages and drawbacks of the proposed strategy. I have only one concern left regarding the α -1,3-glucan compensation in YG medium. It is unclear why the result of α -1,3-glucan labeling with GFP was not shown in the manuscript. Author should add such data in the revised manuscript.

Moreover, if re-formation of α -1,3-glucan was the reason for the pellet in YG medium. This result suggested that other factors, especially the medium composition, may challenge the proposed strategy. Beauvais has previously generated a triple *A. fumigatus* deletion strain that had no α -1,3-glucan at all. And there should be no compensation of α -1,3-glucan in this strain. I would suggest the authors to further construct a new mutant strain based on this triple deletion strain. This would further warranty the compatibility of the proposed strategy.

We have added images of mycelial cells labeled with AGBD–GFP. The AfS35 strain was clearly labeled with AGBD–GFP along the outline of the cells. The Δ *ags1* Δ *gtb3* strain cultured in YG medium (pellet formed) was clearly labeled with AGBD–GFP in the septa and along the hyphal outline. These results suggest that hyphal pellet formation in Δ *ags1* Δ *gtb3* cultured in YG medium depended on α -1,3-glucan synthesized by *ags2* and/or *ags3*. We regret not having constructed the mutant with a triple deletion of α -1,3-glucan synthase genes because of the need for several transformations and the tight revision schedule, but we have labeled the *Aspergillus oryzae* Δ *agsA* Δ *agsB* Δ *agsC* Δ *sphZ* Δ *ugeZ* (AG Δ -GAG Δ) strain with AGBD–GFP. Both the YG- and AMM-cultured AG Δ -GAG Δ cells were scarcely labeled. In both media, AG Δ -GAG Δ hyphae were dispersed. These results support the assumption that the induction of α -1,3-glucan synthesis contributes to pellet formation in YG medium. We have revised the text in lines 220–223 in the

Materials and Methods, lines 318–325 in the Results, and lines 386–390 in the Discussion, and added Figure S9.

Reviewer #3 (Comments for the Author):

First, I am an additional reviewer after the first review. The manuscript entitled "Quantitative monitoring of the mycelial growth of *Aspergillus fumigatus* in liquid culture by optical density" written by Miyazawa K et al. describes a method to monitor hyphal growth of pathogenic filamentous fungus, *A. fumigatus*, showing age1 and gtb3 double mutant using absorbance OD600 as an indicator. This is also the first study to demonstrate that Gtb3 is involved in biosynthesizing glycosaminoglycans (GAGs). The method was used to assess the effects of antifungal drugs used in clinical treatment, and the results were consistent with those based on the method described in CLSI M38-A2. Monitoring growth by absorbance allows for rapid screening of antifungal drugs. Additionally, this study was robustly conducted and had no technical problems. However, I think this study needs to be revised on several points before publication.

1. I feel uncomfortable with the choice of the word "biomass." I think the term is inappropriate to describe the tiny weight of fungus.

In response to your suggestion, we have replaced "biomass" with "mycelial weight" (lines 30, 80, 304, 305, 308, and 381).

2. The cell wall of age1 and gtb3 double mutant may be thinner than the parental strain, making it easier for antifungal agents to penetrate it. Considering such influence, I think the number of antifungal agents' examples in the experiment is too small. Therefore, please present additional data that have been widely verified, such as the effects of chemical compounds (except the antifungal agents shown here) and the effects of some stress conditions (temperature, osmotic pressure and pH etc).

We have evaluated the growth of the *A. fumigatus* AfS35, Δ ags1, Δ gtb3, and

Δags1Δgtb3 strains under stress. The growth was similar among the four strains under several temperature, osmotic stress, and pH conditions. Congo red and calcofluor white are effective inhibitors of the growth of α-1,3-glucan- and/or GAG-deficient strains, which are consistent with our previous reports in *A. nidulans* and *A. oryzae* (Yoshimi et al., PloS One, 2013, doi:10.1371/journal.pone.0054893; Yoshimi et al., J. Appl. Glycobiol., 2017, doi:10.5458/jag.jag.JAG-2017_004; Miyazawa et al., Front. Microbiol., 2019, doi:10.3389/fmicb.2019.02090). We have added these results as Figure S5 and revised the text (lines 97–104 in the Materials and Methods and lines 256–267 in the Results).

3. In the discussion section (line 407), you have said that "establishing culture conditions that prevent pellet formation helps screen antifungal drugs using clinical isolate strain," but this has nothing to do with your findings. Rather, if inhibitors of alpha-glucan and GAG biosynthesis are discovered, you could perform the similar experiment with the inhibitor on a clinical strain. Therefore, I think you should discuss this point.

We have revised the text (lines 430–433 in the Discussion).

November 17, 2021

Dr. Yoshitsugu Miyazaki
National Institute of Infectious Diseases
Department of Chemotherapy and Mycoses
Tokyo 162-8640
Japan

Re: Spectrum00063-21R2 (Quantitative monitoring of mycelial growth of *Aspergillus fumigatus* in liquid culture by optical density)

Dear Dr. Yoshitsugu Miyazaki:

Your manuscript has been accepted, and I am forwarding it to the ASM Journals Department for publication. You will be notified when your proofs are ready to be viewed.

Sincerely,

Slavena Vylkova
Editor, Microbiology Spectrum